# Amyloid-Beta Peptides Trigger Aggregation of Alpha-Synuclein In Vitro

**DOI:** 10.3390/molecules25030580

**Published:** 2020-01-29

**Authors:** Janett Köppen, Anja Schulze, Lisa Machner, Michael Wermann, Rico Eichentopf, Max Guthardt, Angelika Hähnel, Jessica Klehm, Marie-Christin Kriegeskorte, Maike Hartlage-Rübsamen, Markus Morawski, Stephan von Hörsten, Hans-Ulrich Demuth, Steffen Roßner, Stephan Schilling

**Affiliations:** 1Fraunhofer Institute of Cell Therapy and Immunology, Department of Drug Design and Target Validation IZI-MWT, 06120 Halle, Germany; janett.koeppen@izi.fraunhofer.de (J.K.); lisa.machner@uk-halle.de (L.M.); michael.wermann@izi.fraunhofer.de (M.W.); rico.eichentopf@cbp.fraunhofer.de (R.E.); hans-ulrich.demuth@izi.fraunhofer.de (H.-U.D.); stephan.schilling@izi.fraunhofer.de (S.S.); 2Fraunhofer Institute of Cell Therapy and Immunology IZI, 04103 Leipzig, Germany; max.guthardt@izi.fraunhofer.de; 3Fraunhofer Institute for Microstructure of Materials and Systems IMWS, 06120 Halle, Germany; angelika.haehnel@imws.fraunhofer.de (A.H.); jessica.klehm@imws.fraunhofer.de (J.K.); 4Paul Flechsig Institute of Brain Research, University of Leipzig, 04109 Leipzig, Germany; Marie-Christin.Kriegeskorte@medizin.uni-leipzig.de (M.-C.K.); maikerbs@uni-leipzig.de (M.H.-R.); markus.morawski@medizin.uni-leipzig.de (M.M.); steffen.rossner@medizin.uni-leipzig.de (S.R.); 5Friedrich-Alexander-University Erlangen-Nürnberg, Preclinical Experimental Center, 91054 Erlangen, Germany; stephan.v.hoersten@uk-erlangen.de

**Keywords:** alpha-synuclein (α-synuclein), Alzheimer’s disease, amyloid-beta (Aβ), dementia with Lewy bodies, Parkinson’s disease

## Abstract

Alzheimer’s disease (AD) and Parkinson’s disease (PD), including dementia with Lewy bodies (DLB), account for the majority of dementia cases worldwide. Interestingly, a significant number of patients have clinical and neuropathological features of both AD and PD, i.e., the presence of amyloid deposits and Lewy bodies in the neocortex. The identification of α-synuclein peptides in amyloid plaques in DLB brain led to the hypothesis that both peptides mutually interact with each other to facilitate neurodegeneration. In this article, we report the influence of Aβ(1–42) and pGlu-Aβ(3–42) on the aggregation of α-synuclein in vitro. The aggregation of human recombinant α-synuclein was investigated using thioflavin-T fluorescence assay. Fibrils were investigated by means of antibody conjugated immunogold followed by transmission electron microscopy (TEM). Our data demonstrate a significantly increased aggregation propensity of α-synuclein in the presence of minor concentrations of Aβ(1–42) and pGlu-Aβ(3–42) for the first time, but without effect on toxicity on mouse primary neurons. The analysis of the composition of the fibrils by TEM combined with immunogold labeling of the peptides revealed an interaction of α-synuclein and Aβ in vitro, leading to an accelerated fibril formation. The analysis of kinetic data suggests that significantly enhanced nucleus formation accounts for this effect. Additionally, co-occurrence of α-synuclein and Aβ and pGlu-Aβ, respectively, under pathological conditions was confirmed in vivo by double immunofluorescent labelings in brains of aged transgenic mice with amyloid pathology. These observations imply a cross-talk of the amyloid peptides α-synuclein and Aβ species in neurodegeneration. Such effects might be responsible for the co-occurrence of Lewy bodies and plaques in many dementia cases.

## 1. Introduction

Parkinson’s disease (PD) affects 10 million people worldwide and thus represents the second most frequent progressive neurodegenerative disorder, surpassed only by Alzheimer’s disease (AD) [1]. The fibril formation of amyloidogenic proteins is a characteristic feature of both diseases, leading to progressive neuronal death [2]. 

Interestingly, the pathological characteristics of AD and PD show intriguing overlap, e.g., in addition to the typical Lewy bodies, PD patients often display amyloid plaques, a characteristic feature of AD [3]. On the other hand, Lewy bodies have been observed in 32–57% of sporadic AD cases [4,5]. Biochemical analysis showed that these deposits mainly consist of aggregated α-synuclein [6]. Furthermore, the aggregation behavior of α-synuclein underlays the same mechanism as Aβ, describing a sigmoidal kinetic curve as a consequence of primary nucleation [7]. In addition to this process, secondary nucleation is also possible through the combination of both monomers and preformed fibrils [8,9]. 

α-synuclein is ubiquitously expressed in several tissues of the human body with the highest expression level in the central nervous system, where it is especially enriched in pre-synaptic nerve terminals [10]. Although α-synuclein belongs to the group of intrinsically disordered proteins, the soluble peptide has the ability to adopt different conformations, which lead to a multitude of cytosolic, membrane-bound and aggregated states while interacting with ligands or proteins [11,12]. It is proposed that aggregates of α-synuclein can promote misfolding and aggregation of other proteins through cross-seeding, which might be associated with the co-occurrence of multiple neurodegenerative diseases [13,14]. In addition, many post-translational protein modifications, e.g., phosphorylation at S129 of α-synuclein, may also lead to altered aggregation behavior [15]. Mutation studies have shown that the region crucial for toxicity is the non-Aβ-component (NAC) domain, which was also found in the amino acid sequence of Aβ deposits [16]. As shown in circular dichroism spectroscopy studies, this central sequence (aa 61–95) is essential for the formation of β-sheets of α-synuclein [12,17,18,19,20,21,22,23]. The NAC region of α-synuclein was found in Aβ deposits in AD patients, causing dementia with Lewy body (DLB) disease, and both peptides interact with each other through this region [24]. In addition, co-aggregation studies confirmed the mutual influence of the aggregation of Aβ and α-synuclein by stabilizing the formation of aggregates [25]. By means of nuclear magnetic resonance (NMR) spectroscopy, it was shown that α-synuclein and Aβ interact via the synaptic membrane and that rupture of the membrane causes α-synuclein release into the extracellular space [26]. Finally, the interaction of Aβ with α-synuclein causes conformational changes generating NAC fragments.

To elucidate potential cross-connections between AD and PD, we aim at investigating how different amyloidogenic peptides might influence each other to force aggregation and deposition and, eventually, neurodegeneration. New insights into the co-aggregation are also intended to generate a basis for the development of specific antibodies to label heterogeneous aggregates of amyloid peptides typical for different clinical entities. These could help us to gain further understanding of the underlying pathological mechanisms, leading to the development of new therapeutic and diagnostic tools. 

## 2. Results

### 2.1. Expression and Purification of α-Synuclein Variants

In order to investigate the influence of the commonly used His_6_-tag on the aggregation properties of α-synuclein, two different expression strategies were implemented to isolate α-synuclein containing either a His_6_-tag or a cleavable glutathione S-transferase (GST)-tag. Tobacco Etch Virus (TEV) protease was used for cleavage of the fusion part, leaving a native N-terminus. The resulting wt-α-synuclein and the His_6_-tagged variant were used to analyze a possible influence of the tag on the aggregation properties. In a typical approach, we isolated 2 mg/L of His_6_-α-synuclein and 40 mg/L GST-His_6_-α-synuclein. After TEV cleavage of GST-His_6_-α-synuclein, we obtained about 4 mg/l wt-α-synuclein protein. An analytical SDS-PAGE characterizing the purity of GST-His_6_-α-synuclein before cleavage with TEV protease and purified tag-free wt-α-synuclein is shown in Figure 1. The molecular mass of wt-α-synuclein on SDS-PAGE and the corresponding signal in Western blot analysis differs from the theoretical value of 14 kDa due to low binding of SDS by the highly acidic C-terminal sequence of α-synuclein [27]. An additional protein product of ~32 kDa of free GST-tag is visible in lane 1 due to unspecific cleavage by endogenous proteases.

### 2.2. Aggregation of His_6_-α-Synuclein and wt-α-Synuclein

To characterize the in vitro aggregation of His_6_-α-synuclein and wt-α-synuclein at various concentrations, the ThT assay was performed. Fibril formation of both α-synuclein peptides (Appendix A) displayed a typical sigmoidal aggregation behavior [14,28] with decreasing lag phase at increasing α-synuclein concentrations. Nevertheless, prominent discrepancies in aggregation velocity can be observed between the two α-synuclein variants. For example, the nucleation of His_6_-α-synuclein at 75 µM required an average lag time of 111 h compared to 9 h lag time of wt-α-synuclein at the same concentration (Figure 2). Thus, His_6_-α-synuclein showed an approximately 92% longer lag phase than α-synuclein with a native N-terminus. In this regard, Huang et al. proved that kinetics of fibril formation of α-synuclein are concentration-dependent, corroborating the results of our comparative studies [27].

### 2.3. (Co)-aggregation of His_6_-α-Synuclein and wt-α-Synuclein with Aβ(1–42) and pGlu-Aβ(3–42)

To evaluate the effect of Aβ(1–42) and pGlu-Aβ(3–42) on the nucleation process, the α-synuclein variants were analyzed in the presence of Aβ species at pH 7.0. The measurement of ThT binding to amyloid fibrils revealed that at the end of the growth phase and beginning of the steady-state phase, aggregation dynamics of preparations that solely contained α-synuclein peptide variants differed significantly from α-synuclein preparations after addition of Aβ species (Figure 3A,B (left)). However, differences in ThT fluorescence intensity do not necessarily result from different fibril concentration, but could simply arise from two distinct ThT fibril binding modes [29]. Addition of either Aβ species to each of the two α-synuclein peptides had a significant effect on aggregation propensity (Figure 3A,B (right)). Intriguingly, lag phases of wt-α-synuclein are 80% shorter in the presence of Aβ(1–42) and pGlu-Aβ(3–42) (wt-α-synuclein: 18 h, wt-α-synuclein with Aβ(1–42): 2 h, wt-α-synuclein with pGlu-Aβ(3–42): 4 h). In contrast, aggregation kinetics of His_6_-α-synuclein with the addition of Aβ species only show lag phases shortened by about 50% (His_6_-α-synuclein: 87 h, His_6_-α-synuclein with Aβ(1–42): 42 h, His_6_-α-synuclein with pGlu-Aβ(3–42): 35 h). However, the nature of the Aβ species Aβ(1–42) and pGlu-Aβ(3–42), respectively, had no influence on the duration of the nucleation phase. Due to the impaired aggregation kinetics of His_6_-α-synuclein, we focused the following experiments on wt-α-synuclein. 

The co-aggregation of α-synuclein with Aβ(1–42) and pGlu-Aβ(3–42) peptides in vitro was demonstrated by immunogold labeling of Aβ peptides (20 nm gold particle) and wt-α-synuclein aggregates (5 nm gold particles, Figure 4A). Furthermore, double immunofluorescent labelings with specific antibodies directed against the respective Aβ peptides as well as α-synuclein demonstrated co-occurrence in brains of APP-transgenic mice in vivo (Figure 4B). While α-synuclein does not aggregate in wild type mouse brain (not shown), the marked and spatially restricted deposition of α-synuclein around amyloid plaques in Tg2576 mouse brain supports in vitro data on Aβ/α-synuclein protein co-aggregation. This co-labeling pattern was consistently detected irrespective of the brain region with amyloid plaques (hippocampus and neocortex) and of plaque size. For double immunohistochemical labelings in brain sections described above, control experiments in the absence of primary antibodies were carried out. In each case, this resulted in unstained brain sections (not shown). In addition, switching the fluorescent labels of the secondary antibodies (i.e., detection of α-synuclein by secondary donkey anti-rabbit-Cy2 and visualization of Aβ by donkey anti-mouse-Cy3) generated similar results as the procedure outlined above (not shown).

Comparative investigations on the fibril morphologies were performed by TEM. Electron micrographs of stained fibrils indicate the occurrence of different morphologies of wt-α-synuclein fibrils alone and in presence of Aβ(1–42) or pGlu-Aβ(3–42) (Figure 5A). While fibrils of wt-α-synuclein appeared as evenly distributed short, protofibril-like structures, samples from aggregation reactions of wt-α-synuclein with Aβ peptides exhibit networks of long, mature, and branched fibrils. To quantitatively analyze the morphological features of the fibrils, the distribution of fibril width was measured (Figure 5B). The incidence of thinner fibrils is increased for wt-α-synuclein compared to fibrils nucleated by the Aβ species, which show a higher abundance of thicker fibrils. This observation is also reflected by the mean values of fibril diameter of around 10 and 12 nm for wt-α-synuclein and wt-α-synuclein with Aβ peptides, respectively. Statistical analyses of fibril diameters reveal significant differences between fibrils of wt-α-synuclein alone and in the presence of Aβ peptides (Figure 5C).

### 2.4. Toxicity of Co-aggregates of wt-α-Synuclein with Aβ(1–42) and pGlu-Aβ(3–42)

To study the toxicity of wt-α-synuclein and co-aggregates of wt-α-synuclein and Aβ(1–42) or pGlu-Aβ(3–42), an MTT assay was performed using primary neurons from whole mouse brains. The optimal α-synuclein concentration was obtained from concentration-dependent kinetics of fibril formation of wt-α-synuclein at 20–100 µM (Figure 2). No significant cytotoxic potential of 1 µM Aβ(1–42) or pGlu-Aβ(3–42) was observed, which is in accordance with previous studies [30]. However, wt-α-synuclein, as well as co-aggregates of wt-α-synuclein and Aβ(1–42) or pGlu-Aβ(3–42), reduced cell viability significantly at an average to 73% (wt-α-synuclein: 75.8%, wt-α-synuclein with Aβ(1–42): 72.5%, wt-α-synuclein with pGlu-Aβ(3–42): 70.9%) of that of the vehicle control (Figure 6).

## 3. Discussion

α-Synuclein has been identified as the causative protein in the pathogenesis of PD. One major neuropathological hallmark of the disease, besides the degeneration of dopaminergic neurons of the *substantia nigra pars compacta*, are the so-called Lewy bodies that contribute to the neurodegeneration. The triggers for the aggregation have not yet been determined with certainty, but it has been shown that various factors such as metal ions [31], oxidative substances [32], environmental factors, and genetic factors [33,34] can promote aggregation.

In addition to Lewy bodies, deposits of Aβ peptides can be detected in PD brains [3], which implies a mutual influence of these peptides in their aggregation behavior. 

### 3.1. Acceleration of Fibril Formation of α-Synuclein by Aβ(1–42) and pGlu-Aβ(3–42)

To characterize α-synuclein aggregation, it was recombinantly expressed, purified, and analyzed in aggregation experiments. Subsequently, co-aggregation studies were performed in the presence of Aβ peptides. In addition to the already established construct His_6_-α-synuclein, the GST-α-synuclein was produced containing a cleavage site for the TEV protease to generate the native wt-α-synuclein N-terminus. 

Interestingly, comparative investigations of the aggregation kinetics of His_6_-α-synuclein and wt-α-synuclein showed, for the first time, an adverse effect of the polyhistidine-tag on the aggregation efficiency. Although the lag phase decreased by increasing α-synuclein concentrations with both peptide variants, the nucleation of untagged α-synuclein proceeded much faster than that of His_6_-α-synuclein. This effect implies that the length of the lag phase and the aggregation rate is not only dependent on factors such as protein concentration, pH, temperature, and seeding [35,36,37], but also on minor modifications of the amino acid sequence. Thus, working with native recombinant peptides should be considered in future experiments. Similar observations were made in co-aggregation experiments. While aggregation kinetics of both α-synuclein variants showed a sigmoidal, characteristic bipartite curve, the interaction of His_6_-α-synuclein with Aβ(1–42) or pGlu-Aβ(3–42) appeared to be impaired by the His_6_-tag. However, co-aggregation with the Aβ peptides significantly reduced the lag time of both α-synuclein variants, whereas the aggregation rate remained unaffected (Appendix A). This result indicates that the Aβ peptides accelerate the formation of heterogeneous fibrillation nuclei caused by additional nucleation seeds, while not increasing the efficiency of the polymerization process. Previous in vivo studies also demonstrated that the production of α-synuclein oligomers and higher polymers are accelerated in the presence of Aβ(1–42) compared to the aggregation of α-synuclein alone [25,38]. Similar in vitro cross-seeding effects have been reported by Ono et al., where aggregates of Aβ and α-synuclein acted as seeds and promoted the aggregation of each other [39]. However, studies of Chia et al. show that in the presence of monomeric α-synuclein, the secondary nucleation pathway is impeded, resulting in a decreased rate of the formation of Aβ42 aggregates with limited effects on primary pathways [40]. Since α-synuclein fibrils generated an opposite effect, they concluded that the influence of α-synuclein on Aβ42 aggregation depends strongly on the conformational state of α-synuclein. Monomers of α-synuclein possibly bind to Aβ fibrils, thereby suppressing surface-catalyzed secondary nucleation [40].

Since lag phases of α-synuclein are reduced in the presence of Aβ species, we conclude that the primary nucleation pathway is accelerated. We could not observe a significant impact on the aggregation rate (Appendix A). Additional experiments with preformed α-synuclein fibrils would be needed to clear a possible inhibition of the secondary pathway.

The α-synuclein/Aβ co-aggregate formation was visualized by TEM. The immunogold labeling of the in vitro formed fibrils indicated that both α-synuclein and the Aβ peptides are heterogeneously assembled co-aggregates. From the images, it is not clear whether the Aβ peptides form seed nuclei in which both α-synuclein monomers and α-synuclein filaments attach, or whether the Aβ peptides participate passively in the aggregation process of α-synuclein, e.g., through increased oxidative stress [25]. However, protein aggregation may be promoted by synergistic cross-amyloid interactions of monomeric/oligomeric/fibrillar Aβ and α-synuclein driven by hydrophobic amino acids [41]. It is postulated that α-synuclein and Aβ peptides form complexes mediated by direct fibrillogenic interactions between these peptides probably via the highly hydrophobic NAC region of α-synuclein and the hydrophobic regions around residues 16–21 and 29–35 of Aβ [18,24,26,41]. A similar coexistence has been reported for tau and α-synuclein, i.e., abnormal tau aggregates accumulate in numerous cases of α-synuclein deposition and vice versa [42,43,44]. This co-localization indicates a strong interaction between tauopathies and synucleinopathies, which is in strong support of a cross-seeding of pathogenic proteins [43,45,46]. This mutual influence on protein folding contributes to the progression of the disease disrupting cytoskeletal organization, impairing axonal transport, and compromising synaptic organization [38,46,47,48,49,50,51,52,53,54].

The co-appearance of α-synuclein and Aβ was further corroborated by double immunofluorescent labelings of the respective Aβ peptides with α-synuclein in brains of transgenic mice with amyloid pathology in vivo. The pronounced and concentrated deposition of α-synuclein around amyloid plaques supports in vitro data on Aβ/α-synuclein protein co-aggregation. Interestingly, we have recently demonstrated that endogenous mouse huntingtin also co-aggregates to Aβ plaques in Tg2576 mouse cortex and hippocampus [55]. From a more general perspective, this is consistent with the co-aggregation of proteins characteristic for different clinical entities (AD and PD, as well as AD and Huntington’s disease) suggesting cross-disease mechanisms of pathogenic protein co-aggregation events.

The aggregation of α-synuclein may result in various products, including fibrils, soluble oligomers, or insoluble amorphous aggregates [56]. In the TEM images obtained in our study, mainly fibrillar structures with different morphology were detected. Nielsen et al. showed by various biophysical experiments that α-synuclein fibrils tend to form networks and elongated attachments after setting the steady-state phase [57]. Thus, this phase is not exclusively subject to the dynamic balance between degradation and aggregation, but also causes morphological differences of the fibrils. This property of branching is especially observed in fibrils of wt-α-synuclein in the presence of Aβ(1–42) or pGlu-Aβ(3–42). It is striking that fibrils increase in thickness and length after simultaneous incubation with Aβ species, which could be also an indication of a changed fibril structure. Annamalai et al. demonstrated that these polymorphisms also affect Aβ fibrils. They found that in vivo amyloid fibrils of one patient can vary considerably in their three-dimensional architecture, irrespective of amyloid type or source [58]. Polymorphic properties of the fibrils could already be determined from the ThT aggregation assay. When analyzing the aggregation curves, different maximum ThT fluorescence intensities were observed. These could give information about the heterogeneity in the fibril morphology. ThT binds between the β-strands of the aggregates. However, the emitted light may be shielded by the molecular assembly in its environment, or less ThT may be bound due to structural changes in the fibrils. This has already been shown for the Aβ isoforms 1–40 and 1–42 [59].

### 3.2. (Co)-aggregate-mediated Cellular Toxicity

Finally, the toxicity of wt-α-synuclein and co-aggregates thereof with Aβ(1–42) or pGlu-Aβ(3–42) was studied by using primary neurons from whole mouse brains in an MTT assay. Although the results of the ThT assay showed that aggregation velocity is significantly increased in the presence of Aβ(1–42) and pGlu-Aβ(3–42), no increased toxicity was observed compared to treatment of cells with wt-α-synuclein alone. These results suggest that wt-α-synuclein increases neurotoxicity caused by the accumulation of α-synuclein aggregates. Toxicity strongly depends on the conformation of the peptide. Several studies have proven that oligomers and protofibrils, i.e., β-sheet rich structures of Aβ formed during the growth phase, are the toxic species compared to monomers and samples collected during the lag phase [60,61]. The decisive factor for the cytotoxicity is, therefore, the range between the lag phase and steady-state phase of aggregation, which can be analyzed by determining the aggregation rate. Since the aggregation rate remained unaffected (Appendix A), one could postulate that the co-aggregation of α-synuclein with Aβ does not affect the formation of oligomers and protofibrils but accelerates the nucleation process.

Nevertheless, Hoenen et al. postulated that monomers and dimers of α-synuclein might correspond to an early stage of PD and faster progression of the pathology [62]. In fact, they confirmed recent work on the degree of α-synuclein-mediated toxicity by Lashuel et al., with regard to the conformation of the protein [63]. These results are in accordance with the hypothesis that in early stages of PD, small diffusible α-synuclein proteins activate microglia leading to an inflammatory state [64]. Thus, to elucidate the putative influence of Aβ on α-synuclein cytotoxicity, investigation of microglial activation might provide additional insights.

## 4. Materials and Methods 

### 4.1. Cloning, Expression, and Purification of Human α-Synuclein Variants

Cloning of α-synuclein was performed by applying standard cloning procedures using a human cDNA clone purchased from Centic Biotec (Heidelberg, Germany) [65]. The DNA sequence coding for α-synuclein was cloned into a pET28a(+) plasmid vector containing a polyhistidine-tag at the N-terminus. The fusion protein His_6_-α-synuclein was expressed in *E. coli* strain BL21 with 0.4 mM isopropyl β-d-thiogalactosidase (IPTG) at 37 °C for 4 hours. Cell disruption was done by osmotic shock [27]. The first purification step was carried out through Ni^2+^-chelating chromatography on Streamline Chelating resin (Streamline Chelating, GE Healthcare Life Sciences, Uppsala, Sweden) with a cv = 140 mL. The fractions obtained were analyzed and subjected to reversed phase chromatography (Source 15 RPC, GE Healthcare Life Sciences, Uppsala, Sweden), followed by lyophilisation and anion exchange chromatography (MonoQ 5/50GL, GE Healthcare Life Sciences, Uppsala, Sweden). The final buffer used for the experiments was 20 mM Tris/HCl, pH 7.0, containing 100 mM NaCl. The purity of the samples was assessed by SDS-PAGE and mass spectrometry. Protein concentrations were determined using UV absorption at 280 nm. 

To obtain a native *N*-terminus, a second α-synuclein variant (wild-type, wt-α-synuclein) was created. Therefore, the DNA sequence coding for α-synuclein was cloned into a pGEX-4T-1 plasmid vector. The resulting fusion protein contains an *N*-terminal GST-tag, followed by a polyhistidine-tag. The fusion is separated from the α-synuclein sequence by a TEV protease cleavage sequence. TEV protease was isolated as described by Cabrita et al. [66]. The fusion protein GST-His_6_-α-synuclein was expressed in *E. coli* strain BL21 with 0.4 mM IPTG at 21 °C overnight. Cell disruption was done by sonification. A first purification step was carried out through Ni^2+^-chelating chromatography on Streamline Chelating resin (Streamline Chelating, GE Healthcare Life Sciences, Uppsala, Sweden) with a cv = 140 mL. Fractions containing the expression construct were combined, and a second purification step was proceeded with via glutathione sepharose resin and a cv of 19 ml (Glutathione Sepharose 4FF, GE Healthcare Life Sciences, Uppsala, Sweden). The removal of glutathione was achieved by dialysis against buffer containing 100 mM NaCl, 30 mM Tris/HCl pH = 7.6, 0.1 mM DTT, and a membrane with cut-off 6–8 kDa overnight. The TEV cleavage reaction was done at RT overnight, followed by a final purification step via reversed phase chromatography (Source 15 RPC, GE Healthcare Life Sciences, Uppsala, Sweden). Fractions of interest were lyophilized and stored at −20 °C until usage. 

### 4.2. Synthesis of Amyloid Peptides

Aβ(1–42) and pGlu-Aβ(3–42) were synthesized by solid-phase synthesis and purified as described previously [67]. Structures and purities of Aβ peptides were confirmed by mass spectrometry. Before usage, peptides were dissolved in HFIP (Sigma-Aldrich, St. Louis, MO, USA) overnight. The solvents were evaporated under a stream of nitrogen and dissolved in 0.2 M NaOH, followed by buffer and finally titrated with 0.2 M HCl. Recombinant Aβ peptides were obtained as described previously [68].

### 4.3. ThT Fluorescence Assay

The thioflavin T (ThT) assay was carried out essentially as described previously [69]. The assay was conducted in 96-well plates using a FluoStar Optima (BMG Labtech, Ortenberg, Germany) plate reader. The excitation and emission wavelengths were 440 and 490 nm, respectively. For monitoring fibril formation of the α-synuclein variants, 20 µM ThT (Sigma-Aldrich, St. Louis, MO, USA) was added to the aggregation buffer (20 mM Tris/HCl, 100 mM NaCl, pH 7.0). Signals were recorded at 37 °C under continuous shaking (300 rpm) with a time interval of ∼15 min between each recording. For tracking the co-aggregation process of the two α-synuclein variants, Aβ(1–42) or pGlu-Aβ(3–42) were added. To analyze the obtained aggregation curves, we followed the procedure of Hortschansky et al. [70]. Therefore, the lag time (t_lag_) of the aggregation was determined by fitting the straight lines *a* to the baseline of the lag phase and *b* as a tangent to the steepest region of the growth phase curve. t_lag_ is defined as the time point where the two lines *a* and *b* intersect. 

For each peptide, measurements were performed in six cavities of one plate. Obtained data were analyzed using one-way ANOVA, followed by Tukey post-hoc analysis.

### 4.4. Transmission Electron Microscopy and Immunogold Labelling

Fibril samples (5 µL) from an aggregation reaction as described above but without the use of ThT were directly incubated on a formvar carbon-coated copper grid (Plano, Wetzlar, Germany) for 10 min and washed three times with distilled water. Staining was obtained with 2% (*v*/*v*) uranyl acetate (SERVA Electrophoresis GmbH, Heidelberg, Germany) for 5 min. Fibrils were imaged with a LEO EM 912 Omega TEM (Zeiss, Oberkochen, Germany) at 80 kV, and digital micrographs were obtained with a dual-speed 2K-on-axis CCD camera-based YAG scintillator (TRS-Tröndle, Moorenweis, Germany). 

For immunogold labeling, the fibril samples were fixed on the grids with 2% (*w*/*v*) paraformaldehyde (Merck, Darmstadt, Germany) in PBS buffer (Thermo Fisher, Waltham, MA, USA) for 20 min followed by washing with distilled water. The sections were next incubated for 30 min at RT with blocking solution (1% (*w*/*v*) BSA (Bovine Serum Album, Sigma-Aldrich) containing 0.1% Tween-20 (Carl Roth, Karlsruhe, Germany) in PBS). After blocking, the grids were incubated with the first antibody overnight. Depending on the protein, the following antibodies were used: anti-alpha-synuclein filament antibody (rabbit monoclonal, recognizes α-synuclein, MJFR-14-6-4-2, Abcam, Cambridge, UK), 6E10 (mouse monoclonal, recognizes Aβ(1–42), Merck, Darmstadt, Germany) and ME8 (mouse monoclonal, recognizes pGlu-Aβ(3–42), in-house). All primary antibodies were diluted 1:250 in blocking solution. After additional washing with blocking solution, the grids were incubated for 90 min with secondary antibodies coupled to colloidal gold: anti-rabbit IgG 5 nm gold (goat polyclonal, Sigma-Aldrich, St. Louis, MO, USA) and anti-mouse IgG 20 nm gold (goat polyclonal, Abcam, Berlin, Germany), diluted 1:50 in blocking solution. After three washing steps with distilled water, the grids were stained with 2% (*v*/*v*) phosphotungstic acid (Sigma-Aldrich, St. Louis, MO, USA) for 5 min. The imaging of the fibril samples was performed by the use of a TEM/STEM FEI-TecnaiG2 F20 (Hillsboro, OR, USA) in STEM-mode at 200 kV. The electron micrographs were detected using a high-angle annular dark-field detector and finally processed by contrast-inversion.

### 4.5. APP-Transgenic Tg2576 Mice

In this study, APP-transgenic Tg2576 mice developed and described earlier [71] were used as the model for amyloid pathology and to reveal a potential aggregation of α-synuclein to amyloid plaques in vivo. The mice express hAPP695 with the Swedish double mutation (K670N, M671L) as transgene under control of a hamster prion protein promoter. Mice heterozygous for the transgene and wild type littermates are on a mixed C57BL/6 × SJL background. Mice were housed in groups of 3–5 animals per cage and separated by sex, with ad libitum access to water and food with 12 h day/12 h night cycles at 23 °C. The cages contained red plastic houses (Tecniplast) and shredded paper flakes to allow nest building. At the age of six weeks, the transgenicity of the animals was tested by polymerase chain reaction of tail DNA, as described elsewhere [71]. Mice were studied at the age of 18 months. Age-matched non-transgenic littermates served as controls.

### 4.6. Antibodies Against Aβ(1–42), pGlu-Aβ(3–42) and α-Synuclein

To specifically detect Aβ(1–42), pGlu-Aβ(3–42) and α-synuclein in Tg2576 mouse brain sections, we used the mouse monoclonal antibody 6E10 directed against the amino terminus of the Aβ peptide (Millipore; 1:1,000), the pGlu-Aβ-specific mouse monoclonal antibody J8 ([72]; 1:100) and the rabbit monoclonal phospho-Serin129-α-synuclein antibody ab51253 (Abcam, Berlin, Germany, 1:200), which marks aggregated proteins in synucleinopathies [73,74] and can be used as a peripheral prodromal Parkinson’s disease (PD) marker [75]. 

### 4.7. Immunohistochemistry

Tissue preparation. Mice were sacrificed by CO_2_ inhalation until death and transcardially perfused with phosphate-buffered saline (pH 7.4) followed by 4% buffered paraformaldehyde through the left cardiac ventricle. After perfusion fixation, the brain was removed from the skull and placed in the same fixative overnight at 4 °C. Following cryoprotection in 30% sucrose in 0.1 M phosphate buffer for 3 days, coronal sections (30 µm) were cut on a sliding microtome and collected in 0.1 M phosphate buffer containing 0.025% sodium azide.

### 4.8. Double Immunofluorescent Labellings 

Simultaneous immunohistochemical labeling of Aβ and α-synuclein, as well as pGlu-Aβ(3–42) and α-synuclein, was performed in Tg2576 mouse brain sections using a cocktail of the respective mouse monoclonal Aβ antibodies in combination with the rabbit anti-α-synuclein antibody. All sections were pre-treated with 60% methanol for 60 min and unspecific staining was blocked by treatment with TBS containing 5% normal donkey serum and 0.3% Triton-X100 before incubating brain sections with the primary antibody-mix in 5% normal donkey serum and 0.1% Triton-X100 for 24 h at 4 °C. Thereafter, brain sections were washed and transferred to a cocktail of secondary antibodies, i.e., Cy2-conjugated donkey anti-mouse (Dianova, Hamburg, Germany, 1:400), Cy3-conjugated donkey anti-rabbit (Dianova, Hamburg, Germany, 1:400) IgGs in TBS containing 2% BSA for 60 min at RT. Brain sections were then washed, mounted onto glass slides and coverslipped.

### 4.9. Confocal Laser Scanning Microscopy 

Confocal laser scanning microscopy (LSM 880 *fast* Airyscan NLO, Zeiss, Jena, Germany) was performed to reveal co-localization of Aβ(1–42) and of pGlu-Aβ(3–42), respectively, with α-synuclein. The Cy2-labelled Aβ/pGlu-Aβ (green fluorescence) was visualized by excitation at 488 nm and detection of emission at 510 nm using a low-range band pass (505–530 nm) and the Cy3-labelled α-synuclein (red fluorescence) was visualized using excitation at 543 nm and emission at 570 nm. Antibody specificity was confirmed by omitting primary antibodies. Photoshop CS2 (Adobe Systems, Mountain View, CA, USA) was used to process the images obtained by light and confocal laser scanning microscopy with minimal alterations to brightness, sharpness, color saturation, and contrast.

### 4.10. Colorimetric MTT Assay

The toxic effect of wt-α-synuclein alone and in combination with Aβ(1–42) or pGlu-Aβ(3–42) on neurons was assessed using an MTT assay (Sigma-Aldrich, St. Louis, MO, USA). Mouse primary neurons were isolated from whole brains as described by Becker et al. [76], but the embryo brains were removed at E14. Additionally, the cells were triturated with the use of trypsin. On a 24-well plate, 250,000 cells/well were spread and cultured at 37 °C in a humidified atmosphere containing 5% CO_2_. Prior to use, 40 µM wt-α-synuclein in combination with 1 µM Aβ(1–42) or 1 µM pGlu-Aβ(3–42) were pre-incubated at 37 °C for 24 h using cell culture medium. The cultivated cells were then treated with these proteins for 48 hours. Afterwards, MTT (3-[4,5-dimethylthiazol-2-yl]-2,5-diphenyl tetrazolium bromide) was added to the cell medium and incubated for four to six hours. The addition of acidic isopropanol solubilized the insoluble formazan precipitates produced by MTT reduction. The absorbance was determined at 570 nm using a plate reader (Tecan Sunrise, Männedorf, Switzerland). The value directly correlates to the number of viable cells.

## 5. Conclusions

Taken together, these results show, for the first time, an increased aggregation velocity of α-synuclein in the presence of Aβ(1–42) as well as pGlu-Aβ(3–42). Further analyses of the fibrils revealed an interaction of α-synuclein and Aβ in vitro and in vivo, resulting in an accelerated fibril formation most likely caused by an enhanced nucleus formation. These observations suggest a potential cross-talk of different amyloid peptide families such as α-synuclein and Aβ in amyloid-based neurodegenerative diseases. The present results of this study may provide the basis to develop new specific antibodies to label heterogeneous deposits of α-synuclein and Aβ, enabling the development of novel therapeutic and diagnostic approaches.

## Figures and Tables

**Figure 1 molecules-25-00580-f001:**
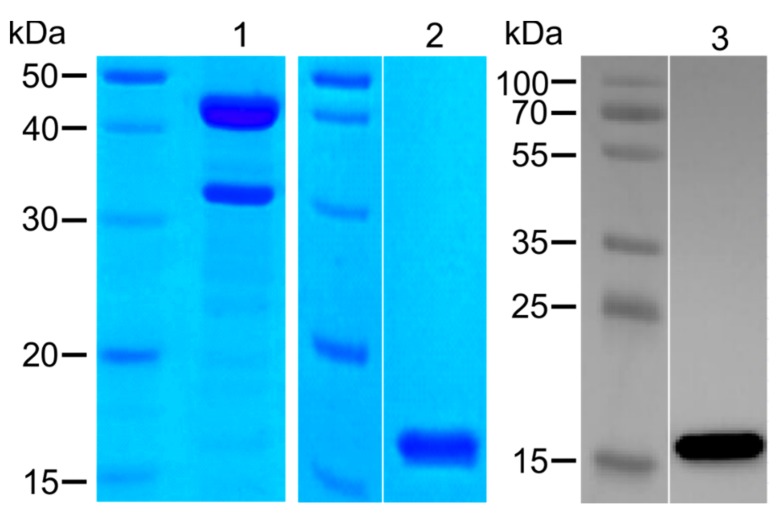
SDS-PAGE and Western blot analysis of α-synuclein. GST-His_6_-α-synuclein and wt-α-synuclein were separated by denaturing SDS-PAGE and stained by Coomassie brilliant blue or by detection using MJFR-antibody (anti-α-synuclein) in Western blot analysis. GST-His_6_-α-synuclein before cleavage (lane 1) migrated as a ~42 kDa monomer. After cleavage with TEV protease and a follow-up purification (lane 2), the resulting wt-α-synuclein showed a molecular mass of about ~15 kDa, which was confirmed by protein-specific antibody (lane 3).

**Figure 2 molecules-25-00580-f002:**
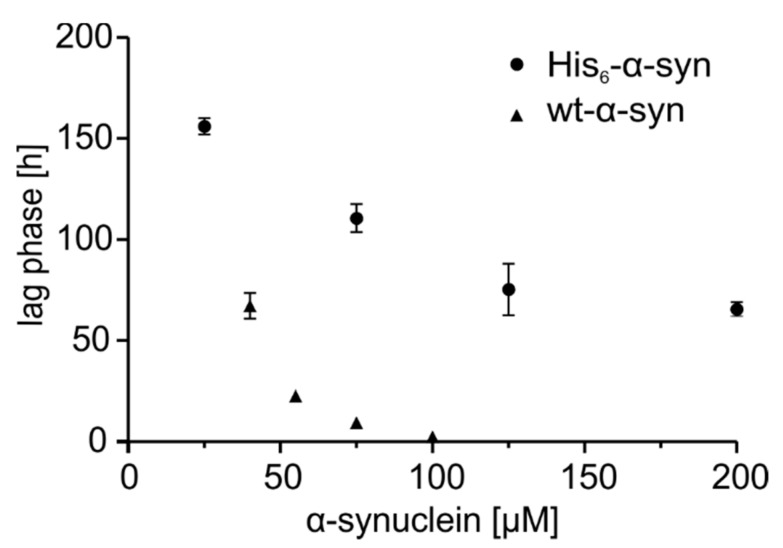
Comparison of lag phase duration obtained from aggregation kinetics of His_6_-α-synuclein (dots, 25 µM, 75 µM, 125 µM, 200 µM) and wt-α-synuclein (triangle, 40 µM, 55 µM, 75 µM, 100 µM) measured by ThT fluorescence at pH 7.0 (*n* = 6).

**Figure 3 molecules-25-00580-f003:**
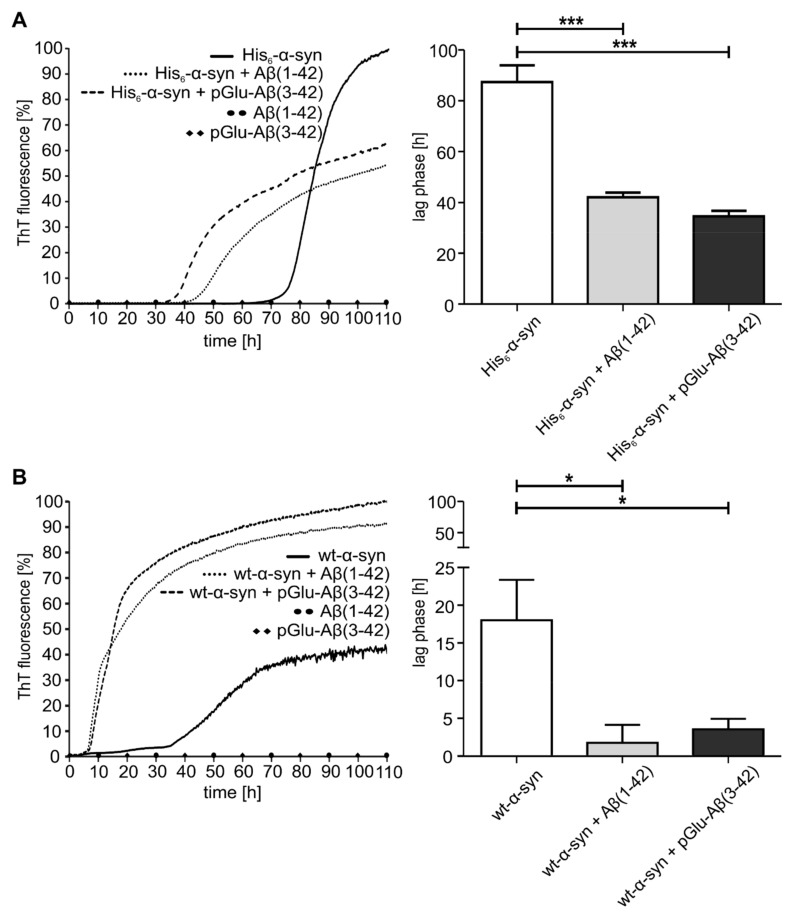
Kinetics of His_6_-α-synuclein and wt-α-synuclein fibril formation and corresponding statistics of lag phase. Fibril formation was induced by incubation of either His_6_-α-synuclein (**A**) or wt-α-synuclein (**B**) assessed by ThT fluorescence at pH 7.0. Seventy-five micromolar of His_6_-α-synuclein or 55 µM wt-α-synuclein were either incubated alone (solid) or in combination with 1 µM Aβ(1–42) (dotted) or 1 µM pGlu-Aβ(3–42) (dashed). Fluorescence intensities of Aβ-peptides alone are visualized as dots. The corresponding statistical analysis of the lag phases was performed as described above (mean ± SD, *n* = 6, * *p* ≤ 0.05 and *** *p* ≤ 0.001, one-way ANOVA and Tukey post-hoc analysis).

**Figure 4 molecules-25-00580-f004:**
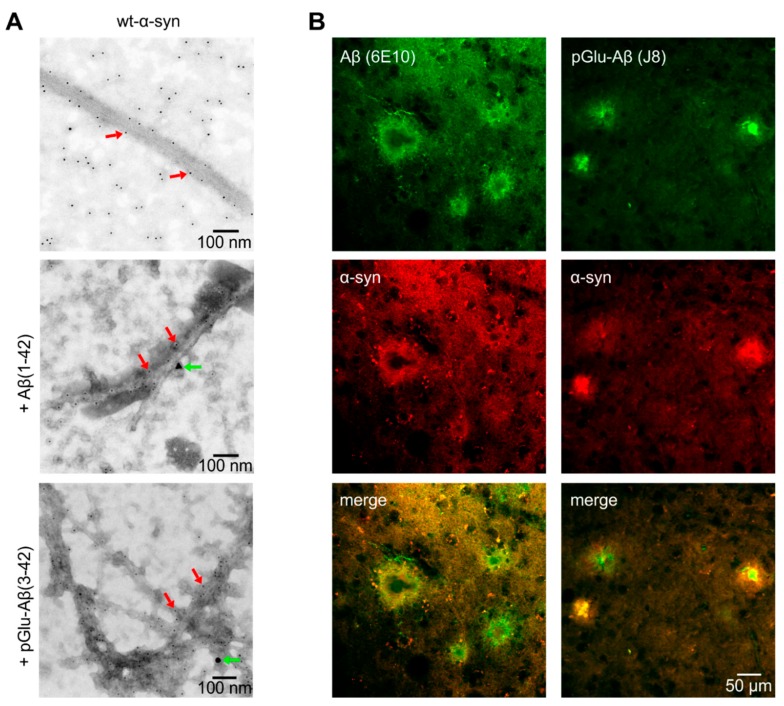
Co-aggregation of wt-α-synuclein with Aβ(1–42) and pGlu-Aβ(3–42) in vitro and in vivo. (**A**) TEM images of amyloid fibrils of wt-α-synuclein alone (top) or in combination with Aβ(1–42) (middle) or pGlu-Aβ(3–42) (bottom). Fibrils were labeled with immunogold particles of defined sizes to identify the different peptides: 5 nm gold particles for the α-synuclein peptides (red arrows) and 20 nm gold particles for the Aβ peptides (green arrows). (**B**) Double immunofluorescent labeling of Aβ (green) and α-synuclein (red) in the parietal cortex of Tg2576 mouse brain. The respective peptides were labeled with mouse monoclonal antibody 6E10 directed against the amino terminus of the Aβ peptide, the pGlu-Aβ-specific mouse monoclonal antibody J8, and the rabbit anti-phospho-Serin129-α-synuclein antibody ab51253, specific for aggregated α-synuclein.

**Figure 5 molecules-25-00580-f005:**
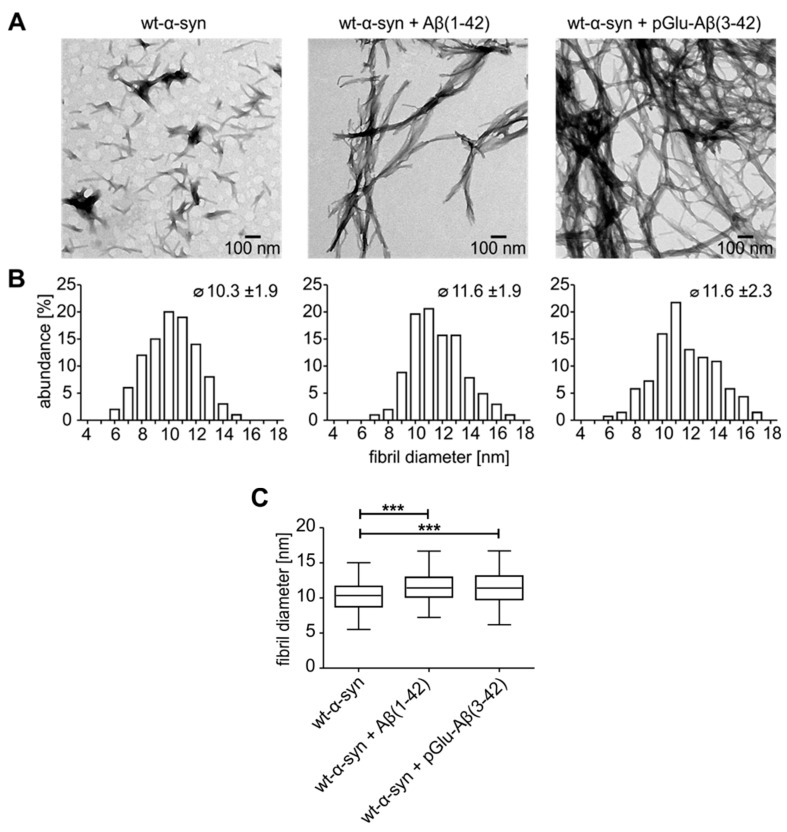
Analysis of amyloid fibrils formed from wt-α-synuclein (75 µM) alone or in the presence of Aβ(1–42) or pGlu-Aβ(3–42) (1 µM). (**A**) Negative stained TEM images and (**B**) corresponding histograms, mean value and standard deviation of fibril diameter quantifications from the above TEM images (*n* = 100). (**C**) Corresponding statistics of fibril diameter (mean ± SD, *n* = 100, *** *p* ≤ 0.001, one-way ANOVA followed by Tukey post-hoc analysis).

**Figure 6 molecules-25-00580-f006:**
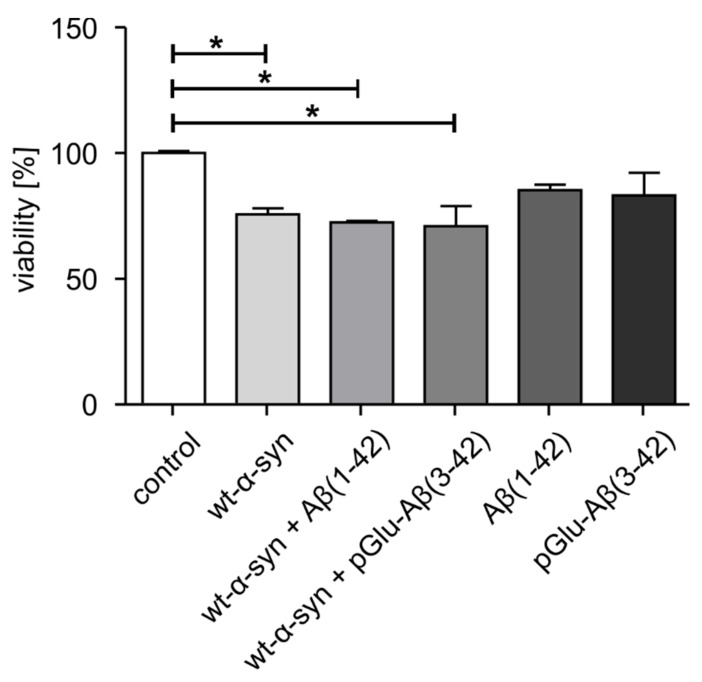
Analysis of cellular toxicity of aggregates of wt-α-synuclein (40 µM) in the presence or absence of 1 µM Aβ(1–42) or 1 µM pGlu-Aβ(3–42). Cell viability was assessed by MTT assay in mouse primary neurons after 72 h of treatment with the peptides (mean ± SD, *n* = 6, * *p* ≤ 0.05, one-way ANOVA followed by Tukey post-hoc analysis).

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
