# Peer review of "Amyloid-Beta Peptides Trigger Aggregation of Alpha-Synuclein In Vitro"

_molecules, 2020, doi:10.3390/molecules25030580_

Round 1

Reviewer 1 Report

This study by Koppen et al., reported the effects of Abeta peptides on alpha-synuclein aggregation. They employed in vitro aggregation assays, as well as excellent microscopic techniques, including TEM. They seem to have well described background, results and discussion. One concern is regarding evaluation of animal models. It is not clear whether signals of alpha-synuclein really reflect aggregated alpha-synuclein or just leakage of signals of Abeta plaques without lacking appropriate controls. Authors should show negative controls, such as normal IgG staining. Also, it is important to confirm whether alpha-synuclein can be biochemically detected in insoluble fraction (e.g., Triton-insoluble, but guanidine-soluble fraction), as most of aggregated Abeta in Tg2576 mice will be detected in such insoluble fraction.

Author Response

Dear Reviewer

We greatly appreciate the contributions of the reviewers to improve the manuscript.

This study by Koeppen et al., reported the effects of Abeta peptides on alpha-synuclein aggregation. They employed in vitro aggregation assays, as well as excellent microscopic techniques, including TEM. They seem to have well described background, results and discussion. One concern is regarding evaluation of animal models. It is not clear whether signals of alpha-synuclein really reflect aggregated alpha-synuclein or just leakage of signals of Abeta plaques without lacking appropriate controls. Authors should show negative controls, such as normal IgG staining. 

Author response: The reviewer asks the important question of appropriate negative controls to confirm the specificity of the immunohistochemical labellings. We regret not having reported these controls in the initially submitted manuscript. Of course, the respective controls are routinely carried out in our laboratory and demonstrate the specificity of the Abeta and α-synuclein labelling. In the revised manuscript, the following controls are reported (page 6, lines 160-167): (i) Immunohistochemistry in the absence of one or both primary antibodies and (ii) switching the label of the secondary antibody from Cy2 to Cy3 and vice versa.

Also, it is important to confirm whether alpha-synuclein can be biochemically detected in insoluble fraction (e.g., Triton-insoluble, but guanidine-soluble fraction), as most of aggregated Abeta in Tg2576 mice will be detected in such insoluble fraction. 

Author response: Biochemically detecting α-synuclein in insoluble fractions is an excellent recommendation that we will take into account in the following experiments. In our experiments (ThT assay, TEM and MTT assay) with recombinant α-synuclein and Aβ, it was shown that the peptides are soluble and require no additional treatment with guanidine hydrochloride. However, working with in vivo materials undoubtedly requires special treatment with guanidine hydrochloride.

Reviewer 2 Report

This paper by Köppen et al., is an interesting study that provide further insights into the influence of Aβ (1-42) and pGlu-31
Aβ(3-42) on the aggregation of α-synuclein in vitro.

Given the shared clinical and neuropathological features between AD and PD that can be found in a fraction of the patients, this study is of potential interest for the community. 

However, before the study can be considered for publication, several points must be addressed:

1) recent studies showed how α-synuclein monomers inhibit Aβ42 aggregation by binding to Aβ42 fibrils, thereby preventing them from catalysing the nucleation of further Aβ42 aggregates.(https://www.pnas.org/content/early/2017/07/10/1700239114)

The authors should provide some explanation for this and contextualise it in their work.

2) The introduction references are somehow outdated. The authors should introduce their study in the context of the most recent literature.

3) LINE 66 the authors should add more up to date references that define the regions relevant for aSyn conformational behaviour (i.e. https://www.ncbi.nlm.nih.gov/pubmed/24871041, etc.)

4) LINE 72 is missing a reference

5) Line 104 Add the "data not shown" in the SI materials

6Line 270 Add the "data not shown" in the SI materials

7) Figure 6, the authors use the MTT to study the toxicity of wt-α-synuclein and of co-aggregates thereof with Aβ(1-42) or pGlu-Aβ(3-260). However, the MTT readout is known for being affected by the nature of the aggregated species. The authors should confirm these results by carrying out an Apoptosis or  Calcium Influx Assay.

8) The authors use a shaking assay at 37C in absence of lipids to monitor the aSyn aggregation. This assays is poorly representing the physiological condition of aSyn aggregation and does not allow to discriminate between primary and secondary processes. The authors should repeat their key aggregation experiments using also an aggregation assays in presence of lipids (https://www.ncbi.nlm.nih.gov/pubmed/25643172) to elucidate the effects of their system on primary nucleation, and use acidic PH conditions with preformed fibrils to elucidate the  contribution of their system on aSyn secondary nucleation (https://www.pnas.org/content/111/21/7671.long)

Author Response

Dear Reviewer

We greatly appreciate the contributions of the reviewers to improve the manuscript.

This paper by Köppen et al., is an interesting study that provide further insights into the influence of Aβ (1-42) and pGlu-Aβ(3-42) on the aggregation of α-synuclein in vitro.

Given the shared clinical and neuropathological features between AD and PD that can be found in a fraction of the patients, this study is of potential interest for the community. 

However, before the study can be considered for publication, several points must be addressed:

1) recent studies showed how α-synuclein monomers inhibit Aβ42 aggregation by binding to Aβ42 fibrils, thereby preventing them from catalysing the nucleation of further Aβ42 aggregates.(https://www.pnas.org/content/early/2017/07/10/1700239114)

The authors should provide some explanation for this and contextualise it in their work.

Authors response: The study of Chia et al. raises an interesting question about the underlying mechanism of the amyloid cascade hypothesis. They show that in the presence of monomeric α-synuclein, the secondary nucleation pathway is impeded, resulting in a decreased rate of the formation of Aβ42 aggregates with limited effects on primary pathways. Since α-synuclein fibrils result in an opposite effect, they conclude that the influence of α-synuclein on Aβ42 aggregation depends strongly on the conformational state of α-synuclein.

Based on the experiments in our manuscript, we can only make a statement on the primary nucleation pathway, which is accelerated as demonstrated by the reduced lag phase of α-synuclein in presence of Aß species. We could not observe a significant impact on the aggregation rate, now added as Supplementary Figure 2. Additional experiments would be needed to clear a possible inhibition of the secondary pathway. We have now discussed this phenomenon based on our data and the recommended publication on page 9, line 245-251.

2) The introduction references are somehow outdated. The authors should introduce their study in the context of the most recent literature.

Authors response: We are grateful for the advice and have improved the introduction based on quotations from recent literature (page 2, lines 58-61 and 65-69).

3) LINE 66 the authors should add more up to date references that define the regions relevant for aSyn conformational behavior (i.e. https://www.ncbi.nlm.nih.gov/pubmed/24871041, etc.)

Authors response: We appreciative this comment and have updated the literature on page 2, lines 67-69).

4) LINE 72 is missing a reference

Authors response: We apologize for the misleading use of the corresponding reference in this section. The paper of Mandal et al. (reference No. 16) provides the information about the interaction between Aβ and α-synuclein via the synaptic membrane but also the generation of NAC fragments caused by this process. We added the reference on page 2, line 80.

5) Line 104 Add the "data not shown" in the SI materials

Authors response: We thank you for the suggestion and modified the text on page 3, line 115 and added the corresponding figure to the supplementary information.

6) Line 270 Add the "data not shown" in the SI materials

Authors response: We thank you for the suggestion and modified the text on page 9, line 239 and page 10, line 310-311 and added the corresponding figure to the supplementary information.

7) Figure 6, the authors use the MTT to study the toxicity of wt-α-synuclein and of co-aggregates thereof with Aβ(1-42) or pGlu-Aβ(3-260). However, the MTT readout is known for being affected by the nature of the aggregated species. The authors should confirm these results by carrying out an Apoptosis or Calcium Influx Assay.

Authors response: The reviewer asks the important question of carrying out alternative methods to MTT like an Apoptosis or Calcium Influx Assay. We thank the reviewer for these excellent recommendation and we will take them into account in the following experiments. However, Iljina et al. (https://pubs.acs.org/doi/pdf/10.1021/acsnano.8b03575) addressed this question by measuring the co-oligomer-induced permeabilization of lipid membranes using a single-vesicle assay that quantifies Ca2+ influx into lipid vesicle due to membrane disruption. They performed the experiments for solutions containing oligomers of wt-α-synuclein or Aβ(1-42) but also co-oligomers of these two species. They report that the co-oligomers contribution to the ability to permeabilize membranes is small. Therefore oligomers of either wt-α-synuclein or Aβ(1-42) alone are significantly more disruptive than in combination.

8) The authors use a shaking assay at 37C in absence of lipids to monitor the aSyn aggregation. This assay is poorly representing the physiological condition of aSyn aggregation and does not allow to discriminate between primary and secondary processes. The authors should repeat their key aggregation experiments using also an aggregation assays in presence of lipids (https://www.ncbi.nlm.nih.gov/pubmed/25643172) to elucidate the effects of their system on primary nucleation, and use acidic PH conditions with preformed fibrils to elucidate the contribution of their system on aSyn secondary nucleation (https://www.pnas.org/content/111/21/7671.long)

Authors response: This is an excellent suggestion raised by the reviewer. Indeed, experiments dealing with secondary nucleation processes induced by seeds or preformed fibrils are scheduled for the next paper. The following publication will specialize on the aggregation behavior of wt-α-synuclein and variants of α-synuclein while the present work deals more with the interaction of wt-α-synuclein and Aβ variants. We have therefore decided to take up this topic in another publication.

Reviewer 3 Report

The authors perform an interesting study by TEM, electronic microscopy, and kinetic analysis, starting from the consolidate evidences that many pathological characteristics are common in Alzheimer Disease (AD) and Parkinson Disease (PD). A Cross-interactions between the AD Amyloid-β Peptide and  α-synuclein protein has been proposed (Wang et al., 2015; Albus et al 2018; Lim et al., 2019; Ke et al., 2017 …etc). But the novelty of this work is represented by the observation that the aggregation of GST-His6-α-synuclein is increased in presence of little concentration of and Aß(1-42) and pGlu-Aß(3-42). In other words amyloid peptides, α-synuclein and Aβ species interacts in the Lewy body formation leading the neurodegeneration.

Only few thinks, should be meliorate:

The authors should explain with more attention that this is the first study focused on Recombinant Human His6-alpha –Synuclein.

The authors should define if the in vitro work was performed in physiological, basic, or acid medium. Should be very interesting to see the changes in kinetic when the pH of the medium is acid.

In figure 1 representing the western blot analysis of α-synuclein, should be important to see the whole gel, to better compare the lines.

Figure 2 is not relevant in the presentation of the result and could be moved in the supplemental material.

For the toxicity study the authors declare to use mouse primary neurons. What kind of primary culture were used: Hippocampal neuron, striatal neuron, mesencephalon neurons? please specify.

Similarly in which brain area were observed the Co-aggregation of wt-α-synuclein with Aβ(1-42) and pGlu-Aβ(3-42)- (figure 4)?

We suggest to characterize the aggregation in cortex versus mesencephalon because there are the areas worst- affected in AD e PD.

To improve the discussion the authors should be consider that there are different works that should be cited: ex: Jinghui Luo et al., Cross-interactions between the Alzheimer Disease Amyloid- Peptide and Other Amyloid Proteins: A Further Aspect of the Amyloid Cascade Hypothesis* Published, JBC Papers in Press, June 20, 2016, DOI 10.1074/jbc.R116.714576; Front. Cell. Neurosci., 18 July 2019 | https://doi.org/10.3389/fncel.2019.00309; Front. Mol. Neurosci., 25 April 2019 | https://doi.org/10.3389/fnmol.2019.00107.

Author Response

Dear Reviewer

We greatly appreciate the contributions of the reviewers to improve the manuscript.

The authors perform an interesting study by TEM, electronic microscopy, and kinetic analysis, starting from the consolidate evidences that many pathological characteristics are common in Alzheimer Disease (AD) and Parkinson Disease (PD). A Cross-interactions between the AD Amyloid-β Peptide and α-synuclein protein has been proposed (Wang et al., 2015; Albus et al 2018; Lim et al., 2019; Ke et al., 2017 …etc). But the novelty of this work is represented by the observation that the aggregation of GST-His6-α-synuclein is increased in presence of little concentration of and Aß(1-42) and pGlu-Aß(3-42). In other words, amyloid peptides, α-synuclein and Aβ species interacts in the Lewy body formation leading the neurodegeneration.

Only few thinks, should be meliorate:

The authors should explain with more attention that this is the first study focused on Recombinant Human His6-alpha –Synuclein.

Authors response: We thank you for the suggestion and modified the text on page 1, line 36 and page 8, line 229 to emphasize the new value of this study.

The authors should define if the in vitro work was performed in physiological, basic, or acid medium. Should be very interesting to see the changes in kinetic when the pH of the medium is acid.

Authors response: The in vitro aggregation measurements were performed at pH 7.0. This information is now included in the captions of figure 2 and 3 and in page 4, line 130. We did not analyze an aggregation of α-synuclein at acidic pH, but Sulatskaya et al. report about α-synuclein fibrillogenesis at pH 3.7 in 0.2 M acetate buffer to enhance the aggregation velocity (https://www.ncbi.nlm.nih.gov/pmc/articles/PMC6163839/#!po=16.6667). They analyzed 1 mg/ml (69 µM) wt-α-synuclein using ThT fluorescence and obtained a lag phase of approximately 8 hours, which is almost identical to the lag phase we determined with 75 µM wt-α-synuclein (9 hours). Nevertheless, at acidic pH, we would assume an even faster aggregation (shorter lag phase).

In figure 1 representing the western blot analysis of α-synuclein, should be important to see the whole gel, to better compare the lines.

Authors response: We apologize for the misleading representation of the SDS-PAGE and Western blot analysis of α-synuclein and replaced figure 1 (line 105) with an unambiguous picture.

Figure 2 is not relevant in the presentation of the result and could be moved in the supplemental material.

Authors response: We decided to show this figure in the main part of the manuscript to underline the concentration-dependence of wt-α-synuclein on the one hand and between wt-α-synuclein and His6-α-synuclein on the other, which is published for the first time.

For the toxicity study the authors declare to use mouse primary neurons. What kind of primary culture were used: Hippocampal neuron, striatal neuron, mesencephalon neurons? please specify.

Authors response: In the MTT assay, we used whole brain neurons. This is now mentioned in the manuscript on page 10, lines 302 and page 13, line 446. Since the brains were taken from embryos ≤ 14 days (E14), we dissected undifferentiated neurons that only differentiate based on the cells in the environment after the subsequent cultivation. In addition, this preparation is largely free of astrocytes and microglia.

Similarly in which brain area were observed the Co-aggregation of wt-α-synuclein with Aβ(1-42) and pGlu-Aβ(3-42)- (figure 4)?

Authors response: The co-aggregation of endogenous α-synuclein and Aβ or pGlu-Aβ(3-42) was consistently observed in the periphery of Aß plaques independent of the brain region and plaque size. This is now mentioned in the manuscript on page 6, lines 161-163.

We suggest to characterize the aggregation in cortex versus mesencephalon because there are the areas worst- affected in AD and PD.

Authors response: This is an interesting point raised by the reviewer based on the predominant pattern of pathologies in AD and PD. However, unlike in the human brain condition, the mesencephalon of Tg2576 mice does unfortunately barely show any amyloid plaques. This makes it impossible to characterize protein co-aggregation in this brain region. In cortex and hippocampus of Tg2576 mice, on the other hand, amyloid plaques are highly abundant and endogenous α-synuclein is expressed, which allows studying protein co-aggregation in these structures.

To improve the discussion the authors should be consider that there are different works that should be cited: ex: Jinghui Luo et al., Cross-interactions between the Alzheimer Disease Amyloid- Peptide and Other Amyloid Proteins: A Further Aspect of the Amyloid Cascade Hypothesis* Published, JBC Papers in Press, June 20, 2016, DOI 10.1074/jbc.R116.714576; Front. Cell. Neurosci., 18 July 2019 | https://doi.org/10.3389/fncel.2019.00309; Front. Mol. Neurosci., 25 April 2019 | https://doi.org/10.3389/fnmol.2019.00107.

Authors response: We are grateful for the literature advice and have improved the discussion based on quotations from the sources mentioned (page 9, lines 245-256; 262-264; 266-272).

Round 2

Reviewer 3 Report

I am satisfied with this version of the manuscript.

I think that now, it is sufficiently improved.